# Specific microRNA Signature Kinetics in *Porphyromonas gingivalis*-Induced Periodontitis

**DOI:** 10.3390/ijms24032327

**Published:** 2023-01-24

**Authors:** Chairmandurai Aravindraja, Krishna Mukesh Vekariya, Ruben Botello-Escalante, Shaik O. Rahaman, Edward K. L. Chan, Lakshmyya Kesavalu

**Affiliations:** 1Department of Periodontology, College of Dentistry, University of Florida, Gainesville, FL 32610, USA; 2Department of Nutrition and Food Science, University of Maryland, College Park, MD 20742, USA; 3Department of Oral Biology, College of Dentistry, University of Florida, Gainesville, FL 32610, USA

**Keywords:** periodontal disease, *Porphyromonas gingivalis*, experimental periodontitis, miRNAs, nanoString analysis, differential expression miRNAs, in vivo oral infection

## Abstract

*Porphyromonas gingivalis* is one of the major bacteria constituting the subgingival pathogenic polymicrobial milieu during periodontitis. Our objective is to determine the global microRNA (miRNA, miR) expression kinetics in 8- and 16-weeks duration of *P. gingivalis* infection in C57BL/6J mice and to identify the miRNA signatures at specific time-points in mice. We evaluated differential expression (DE) miRNAs in mandibles (*n* = 10) using high-throughput NanoString nCounter^®^ miRNA expression panels. The bacterial colonization, alveolar bone resorption (ABR), serum immunoglobulin G (IgG) antibodies, and bacterial dissemination were confirmed. In addition, all the infected mice showed bacterial colonization on the gingival surface, significant increases in ABR (*p* < 0.0001), and specific IgG antibody responses (*p* < 0.05–0.001). The miRNA profiling showed 26 upregulated miRNAs (e.g., miR-804, miR-690) and 14 downregulated miRNAs (e.g., miR-1902, miR-1937a) during an 8-weeks infection, whereas 7 upregulated miRNAs (e.g., miR-145, miR-195) and one downregulated miR-302b were identified during a 16-weeks infection. Both miR-103 and miR-30d were commonly upregulated at both time-points, and all the DE miRNAs were unique to the specific time-points. However, miR-31, miR-125b, miR-15a, and miR-195 observed in *P. gingivalis*-infected mouse mandibles were also identified in the gingival tissues of periodontitis patients. None of the previously identified miRNAs reported in in vitro studies using cell lines (periodontal ligament cells, gingival epithelial cells, human leukemia monocytic cell line (THP-1), and B cells) exposed to *P. gingivalis* lipopolysaccharide were observed in the in vivo study. Most of the pathways (endocytosis, bacterial invasion, and FcR-mediated phagocytosis) targeted by the DE miRNAs were linked with bacterial pathogen recognition and clearance. Further, eighteen miRNAs were closely associated with the bacterial invasion of epithelial cells. This study highlights the altered expression of miRNA in gingiva, and their expression depends on the time-points of infection. This is the first in vivo study that identified specific signature miRNAs (miR-103 and miR-30d) in *P. gingivalis* invasion of epithelial cells, establishes a link between miRNA and development of periodontitis and helping to better understand the pathobiology of periodontitis.

## 1. Introduction

The microRNAs (miRNAs, miR) are a class of small, noncoding regulatory RNA that play an important function in gene regulation/expression in infectious diseases. These miRNAs are critical for normal development, and their aberrant expression is associated with various diseases. Several studies have found significant roles for miRNAs in regulating the innate and adaptive immune responses to microbial infection in tuberculosis, diabetes, rheumatoid arthritis, and atherosclerosis/cardiovascular diseases. The dysregulation of numerous miRNAs plays significant roles in the development of many systemic diseases. There are few in vivo reports that directly link the inflammation-associated miRNAs with periodontal disease (PD) progression [1,2]. The miRNAs serve as alternative genetic inhibitory transcriptional endpoints during infection to maintain periodontal tissue homeostasis via differentiation of periodontal stem cells along with proper functioning of osteoblasts and osteoclasts [3,4]. Further, miRNAs can be used as a potential diagnostic biomarker to identify the disease conditions associated with periodontitis. Our published in vitro and in vivo studies identified a few dominant miRNAs in oral bacteria mediated PD [1,2]. Previously, we reported sex-specific differential miRNA expression in mice infected with partial human mouth microbes (PAHMM) using a novel ecological time-sequential polybacterial periodontal infection (ETSPPI) mouse model. The PAHMM infection model utilized time-sequential infection of mice with five different bacteria: *Streptococcus gordonii* (early colonizer), *Fusobacterium nucleatum* (intermediate colonizer), *Porphyromonas gingivalis*, *Treponema denticola*, and *Tannerella forsythia* (late colonizers). Though we utilized polybacterial infection in the previous study [2], reports on how individual bacteria on the mice’s gingiva mediate miRNA expression at different time points are limited. Furthermore, most of the studies reported the DE (differential expression) of miRNA induction in vitro by exposing cultured cell lines (periodontal ligament cells, gingival epithelial cells, THP-1, and B cells) to LPS, proteins, and nucleic acids found in the membrane or lumen of outer membrane vesicles (OMVs) [5,6]. To the best of our knowledge, no studies have used live *P. gingivalis* to study the DE of altered global miRNA expression in the in vivo periodontitis model. Several miRNAs (e.g., miRNA-128, miRNA-146, miRNA-203, and miRNA-584) identified from in vitro single cell studies do not translate to the miRNA profile with live *P. gingivalis* diseased and healthy periodontal tissues as well as human diseased and healthy tissues, highlighting the importance of our preclinical in vivo studies with live *P. gingivalis*.

The invasive potential of *P. gingivalis* has also been linked with several systemic diseases such as cardiovascular diseases, Alzheimer’s disease, and rheumatoid arthritis [7,8,9]. Hence, studying the miRNA expression kinetics in the *P. gingivalis*-induced experimental periodontitis model provides time-points (8- and 16-weeks) for the expression of altered miRNA and its associated functions. Accordingly, we performed an in vivo *P. gingivalis* infection to evaluate differential mandibular miRNA profiles and miRNA-dependent transcriptional and regulatory phenomena associated with immune signaling events in epithelial cells. Further, this is the first study that used a classical approach in examining altered miRNA global profiles from *P. gingivalis*-diseased mouse gingival tissues at different time-points and sham-infected normal tissues.

We designed and analyzed the global miRNA expression kinetics at two time points (8 and 16 weeks) in *P. gingivalis*-infected male and female C57BL/6J mice. We used novel high-throughput NanoString analysis with nCounter miRNA expression profiling to study the DE kinetics of miRNA after 8- and 16-weeks of *P. gingivalis* infection. We reported bacterial colonization of the gingival surface on all the mice, a significant increase in ABR, and a specific IgG antibody response in the serum of the *P. gingivalis*-infected mice. We also showed specific miRNA signature profiles in 8- and 16-weeks duration of infected and sham-infected healthy mouse mandibles.

## 2. Results

### 2.1. P. gingivalis Gingival Infection Showed Bacterial Colonization, Induced ABR, IgG Antibody Induction and Dissemination to Distal Organs

All mice (*n* = 10) infected with *P. gingivalis* at 8 and 16 weeks showed bacterial colonization on their gingival surface after the second infection cycle. None of the sham-infected mice were positive for the presence of a *P. gingivalis* 16S rRNA gene amplicon (Table 1). Furthermore, gingival infection with *P. gingivalis* significantly induced horizontal ABR. The ABR was significantly higher in the mandible lingual and maxilla palatal (*p* < 0.0001) for both the 8-weeks and 16-weeks duration of *P. gingivalis*-infected mice (Figure 1B,C). Similarly, the serum IgG immune response against *P. gingivalis* was also significantly higher at 8 weeks (*p* < 0.05) and 16 weeks (*p* < 0.0001) (Figure 1D). In addition, minimal *P. gingivalis* dissemination to the lungs, liver, kidneys, and spleen were observed in 16-weeks duration of *P. gingivalis*-infected mice, whereas one female mouse was positive for the bacterial genomic DNA in their heart and brain tissue (Appendix A).

### 2.2. Unique and Common miRNAs in 8- and 16-Week Time-Points of P. gingivalis-Infected Mice Mandibles

Stringent quality checks were made in order to identify the DE miRNAs at 8-weeks and 16-weeks duration of infection in mice. A *p*-value of <0.05 and the fold change of 1.1 were taken into further analysis and considered to be significant. Twenty-six upregulated and 14 downregulated miRNAs were identified in the 8-weeks duration of *P. gingivalis*-infected mice compared to the 8-weeks duration of sham-infected mice. Three upregulated and two downregulated miRNAs were identified between 8-week-old *P. gingivalis*-infected male and female mice. Seven upregulated and one downregulated miRNA were identified in the 16-weeks of *P. gingivalis*-infected mice compared to the 16-weeks of sham-infected mice (Table 2 and Appendix A). Thirty-nine upregulated and nine downregulated miRNAs were identified between 16-weeks duration of *P. gingivalis*-infected male and female mice. The volcano plot analysis depicted 11 upregulated and 37 downregulated miRNAs in 8-weeks of *P. gingivalis*-infected mice compared to the 16-weeks duration of *P. gingivalis*-infected mice (Figure 2A). Two miRNAs, miR-103 and miR-30d, were commonly upregulated in 8- and 16-weeks of infection with *P. gingivalis*, and all the DE miRNAs were unique to the specific time point (Figure 2B). None of the previously identified inflammatory miRNAs (miR-146a, miR-155, and miR-132) were DE at each time point of infection in both male and female mice.

The number of differentially expressed (DE) miRNAs is shown for *P. gingivalis* infected mice after 8- and 16-weeks of infections. The commonly expressed miRNAs between 8-weeks and 16-weeks bacterially infected mice are shown in brackets. Most of the miRNAs expressed in bacterially infected mice were unique and specific to the 8- and 16-week infections.

### 2.3. Predicted Functional Pathway Analysis of DE miRNAs

The predicted functional pathway analysis of the DE miRNAs using KEGG in *P. gingivalis*-infected mouse mandibles at both 8 and 16 weeks identified several pathways, such as TNF (tumor necrosis factor)-signaling. Further, 18 DE miRNAs expressed in 8- and 16-weeks of *P. gingivalis*-infected mice were found to be involved in the bacterial invasion of epithelial cells (Figure 2C and Appendix A). Two miRNAs that were uniquely upregulated in 8- and 16-week *P. gingivalis*-infected mice, miR-103 and miR-30d, were also found to be involved in bacterial invasion of epithelial cells. In addition, other pathways that are linked with infection and host cell associations, such as adherens junctions, endocytosis, TNF signaling, the lysososme pathway, and TGF-β signaling, were also identified (Table 3). 

## 3. Discussion

*P. gingivalis* is one of the dominant bacteria constituting the subgingival pathogenic polymicrobial milieu during severe periodontitis. The subgingival disease microbiome consists of hundreds of microbial species, each expressing various pro-inflammatory factors likely to be involved in PD. Important functions such as inflammatory reactions on the gingival surface and bone and matrix synthesis are maintained by several non-coding small miRNAs [4]. These miRNAs play several roles in complex physiological interactions during PD progression, suggesting the significance of miRNAs in PD. In this study, we focused on how viable *P. gingivalis* selectively modulates host epithelial cell responses with specific miRNA expression during the progression of PD.

Additionally, both male and female mice infected with *P. gingivalis* had colonies on their gingival surfaces after 2 infection cycles, which were continued until 8 and 16 weeks of chronic infection, respectively. The mice infected with *P. gingivalis* showed significantly higher ABR at both time points. Since ABR is considered a periodontal disease outcome measure, the current data directly provide evidence that male and female mice infected with *P. gingivalis* exhibited periodontitis and clearly showed the invasive potential of *P. gingivalis*. No significant difference in ABR was observed between 8-weeks and 16-weeks chronic infection of *P. gingivalis*. Although the current study was designed with a monobacterial infection, the obtained ABR with a *P. gingivalis* infection was similar to the ABR obtained through our previous ETSPPI infection mouse model with a multi-species (PAHMM) polybacterial infection [2]. Similarly, significant levels of serum IgG antibodies were observed in *P. gingivalis*-infected mice at both 8 and 16 weeks of infection. Though IgG levels were significant at both time points, the significance level was higher in 16-weeks infected mice than in 8-weeks infected mice. Higher levels of IgG antibody were induced in the 16-weeks infected mice since the 16-weeks infected mice were infected with eight infection cycles (chronic infection) compared with only four infection cycles in the 8-weeks group. We previously reported robust dissemination of oral bacteria in mice infected with PAHMM using an ETSPPI model. We observe minimal *P. gingivalis* dissemination to distal organs in 16-weeks *P. gingivalis*-infected mice. This indicates that under polybacterial infection, bacteria demonstrate physiological, nutritional, and metabolic synergistic interactions in the gingiva. This codependence allows the bacteria to grow and invade the oral cavity, leading to systemic intravascular dissemination. Though *P. gingivalis* is highly invasive, the current dissemination data indicate *P. gingivalis* requires early colonizers (e.g., *S. gordonii*), intermediate colonizers (e.g., *F. nucleatum*), and other late colonizers (*T. denticola, T*. *forsythia*) for the robust dissemination to other distal organs. 

The high-throughput miRNA profiling of mouse mandibles infected with *P. gingivalis* showed different miRNA gene signatures in both 8- and 16-week infections. Most of the DE miRNAs were unique and specific to the time point, providing a strong consensus that miRNA expression is transient and time-dependent. Further, the DE miRNAs in male and female mice were unique to their respective time points. This data is in line with our previous reports that the DE miRNAs in male and female mice infected with PAHMM in the ETSPPI mouse model were unique [2]. The DE miRNAs expressed in 8-weeks infected mice were not found in the 16-weeks infected mice. The two miRNAs, miR-103 and miR-30d, were commonly expressed and upregulated at both time points. However, both of the miRNAs are not expressed in human gingival tissues in patients with periodontitis. The miR-103 was shown to be highly expressed in ApoE^-/-^ mice and linked with atherosclerosis [25]. The American Heart Association supports an association between periodontal disease and atherosclerosis [39], and we have established the casual link between *P. gingivalis* and atherosclerosis [40]. The upregulated expression of miR-103 at both time points in *P. gingivalis*-infected mouse mandibles further strengthens the existing link between periodontal disease and atherosclerosis. On the contrary, miR-30d was reported to promote cardiomyocyte pyroptosis in diabetic cardiomyopathy [27], and this supports the bi-directional relationship between PD and diabetes. As both miRNAs are also shown to be upregulated in systemic diseases, these two miRNAs induced in *P. gingivalis* infected mouse mandibles may be considered a potential biomarker for periodontal disease progression and also for other systemic diseases.

The four upregulated miRNAs (miR-31, miR-125b, miR-15a, and miR-195) in *P. gingivalis*-infected mouse mandibles were also identified in gingival tissues of periodontitis patients [41], indicating preclinical in vivo miRs data corroborates with clinical PD miRs data (Table 3). Hence, along with miR-103 and miR-30d, these miRNAs may be considered potential diagnostic biomarkers for PD. The miR-31 was also reported to be expressed in various malignancies such as ovarian, prostrate, glioma, and breast cancer [12]. In addition, miR-125b has been associated with ischemic stroke [22], whereas miR-15a has been shown to be a negative regulator of post-ischemic cerebral angiogenesis and long-term neurological recovery [29]. The miR-195 is reported to modulate cell proliferation in colorectal cancers [36]. Since these miRNAs were identified in the gingival tissues of periodontitis patients and other systemic diseases, further in-depth analysis may reveal the interlinked pathways or shared mRNA targets of these miRNAs in both disease states. Other identified upregulated miRNAs, such as miR-19b, miR-193, miR-322, and miR-301a, have been found to be closely associated with endothelial cell apoptosis in coronary heart disease. In addition, miR-1224, miR-191, and miR-30e have been closely associated with ischemic stroke and neuronal inflammation in intracerebral hemorrhage. Other upregulated miRNAs in this study, such as miR-185, miR-22, miR-152, miR-423, miR-151, miR-28, and miR-145, were also found to be upregulated in various malignancies such as prostrate, breast, glioma, gastric, and hepatocellular carcinoma. This data suggests that bacteria-mediated miRNA expression in gingival tissues and other systemic diseases have some common mRNA targets that mediate disease progression. We have previously reported an upregulated expression of inflammatory miRNAs (miR-146a, miR-132, and miR-155) in ApoE^-/-^ mice infected with late colonizers [1]. Many reports have shown a higher expression of miR-146a in patients with periodontitis than healthy individuals [42]. However, we did not observe upregulation of these three miRNAs in both the 8- and 16-weeks of *P. gingivalis* infection. Similar data was also observed in our previous PAHMM-mediated ETSPII infection. 

The KEGG analysis revealed that most of the signaling pathways targeted by the miRNAs that were DE during *P. gingivalis* at both time-points were linked with bacterial pathogen recognition and clearance. Some pathways linked with these functions were endocytosis, bacterial invasion of cells, and FcR-mediated phagocytosis. Further, 18 miRNAs were found to be closely associated with bacterial invasion of epithelial cells. This is the first study that identified the specific miRNAs involved in *P. gingivalis’* invasion of epithelial cells that could be potentially used as biomarkers for periodontitis. These 18 miRNAs were shown to regulate the gene expression that mediates invasion of epithelial cells. Other pathways linked with infection and host-cell associations, such as adherens junctions, endocytosis, TNF signaling, lysosome- and TGF (transforming growth factor)-β signaling pathways, were also identified. One of the limitations of the current study is the inability to utilize salivary fluids in this microRNA analysis. 

## 4. Materials and Methods

### 4.1. Topical Gingival Infection of P. gingivalis for the Induction of Periodontitis

The *P. gingivalis* strain 381 was grown in Brucella blood agar plates supplemented with hemin and vitamin-K (Hardy Diagnostics, Santa Maria, CA, USA) in a Coy anaerobic chamber for three days. Both male and female C57BL/6J mice were received from Jackson Laboratories (Bar Harbor, ME, USA) and were housed in microisolator cages and fed standard chow and sterile water ad libitum. The sham-infected mice were maintained in separate rooms in order to avoid cross-contamination. All mouse cages were maintained at 25 °C with 12-h alternating periods of light and dark. The mouse infection, sampling, and euthanasia procedures were done in accordance with the approved protocol guidelines from the University of Florida Institutional Animal Care and Use Committee (IACUC protocol #202200000223). Approximately 9–10-week-old male and female mice were randomly divided into four groups (*n* = 10 mice/per group; 5 males and 5 females Group-I: *Pg*-infected-8 weeks; Group-II: *Pg*-infected-16 weeks; Group-III: Sham-infected-8 weeks; Group-IV: Sham-infected-16 weeks). The bacterial infection was performed in each group for 8- and 16-weeks [infection cycle consists of four days per week for every alternate week (Figure 1A)]. Kanamycin (500 mg/mL) was administered in sterile drinking water for three days to suppress the existing oral bacteria, followed by rinsing with 0.12% chlorhexidine gluconate (Peridex: 3M ESPE Dental Products, St. Paul, MN, USA). After an antibiotic washout period, the mice were given a topical gingival infection with *P. gingivalis* (10^8^ cells) suspended in an equal volume of reduced transport medium (RTF) and carboxymethylcellulose (CMC). An equal volume of RTF and CMC was used as a vehicle control for sham-infected mice, as described previously [2,7]. The mice were euthanized at their respective time points, and their mandibles, maxilla, and distal organs-brain, heart, lungs, liver, spleen, and kidneywere collected. This study complied with the ARRIVE guidelines (Animal Research: Reporting In Vivo Experiments).

### 4.2. Identification of P. gingivalis Genomic DNA in Gingival Plaques, Bacterial Dissemination and IgG Antibody Quantification

After each infection, the mouse gingival surface was swabbed with a sterile cotton tip applicator and placed in TE buffer. The *P. gingivalis* genomic DNA was detected using 16S rRNA gene specific primers [New England Biolabs (NEB, Ipswich, MA, USA)] as described previously [2]. In addition, genomic DNA from distal organs such as the heart, lung, brain, liver, kidney, and spleen were extracted, and 16S rRNA gene amplification was done to identify the *P. gingivalis* genomic DNA [2]. The serum IgG antibodies against the bacteria were quantified using an enzyme-linked immunosorbent assay (ELISA), and the concentrations were determined by the gravimetric standard curve [2]. 

### 4.3. Mesaurement of Horizontal Alveolar Bone Resorption (ABR)

The horizontal ABR was measured in the right mandible (lingual) along with the left and right maxilla (buccal and palatal) as described previously [2]. The mice’s jaws were autoclaved, defleshed, and bleached with 3% hydrogen peroxide for 30 min. After drying the samples, two-dimensional imaging was done using a stereo dissecting microscope (Stereo Discovery V8, Carl Zeiss Microimaging, Inc. Thornwood, NY, USA). A line tool was used to measure the horizontal ABR between the cementoenamel junction (CEJ) and the alveolar bone crest (ABC) (AxioVision LE 29A 4.6.3, Thornwood, NY, USA) [2,8]. 

### 4.4. miRNA Expression Kinetics Using NanoString nCounter miRNA Panels

The total RNA from the left mandibles (*n* = 10 mice per group; 5 males and 5 females) of bacterially and sham-infected mice was extracted using the miRVana Isolation Kit (Ambion, Austin, TX, USA) [2]. In addition, the left mandibles from each mouse were homogenized using the handheld rotor-stator homogenizer with sterile individual TissueRuptor disposable probes (Qiagen; Germantown, MD, USA) for each specimen. After homogenization, each specimen was lysed in a denaturing lysis solution that stabilized RNAs and inactivated RNases. The lysed samples were subjected to an acid-phenol:chloroform extraction that removed all the cellular components (protein, DNA, and other cellular products). The aqueous phase was removed, transferred to the fresh nuclease-free microcentrifuge tube, and 1.25 volumes of 100% ethanol were added. This mixture was transferred to the filter cartridge placed into the collection tube and centrifuged for 15 s at 10,000× *g*. The filter cartridge was washed with Wash Solution-1, followed by Wash Solution-2/3. After washing, the total RNA was eluted from the filter cartridge with nuclease-free water. The RNA concentration and purity were determined using a Take3 micro-volume plate in an Epoch Microplate Spectrophotometer (BioTek, USA; Winooski, VT, USA), and RNA quantification was performed in technical duplicates for each sample. The high-quality RNA samples (100 ng) with an OD 260/230 ratio of >1.8 and an OD 260/280 ratio of >2 were taken for NanoString analysis. miRNA expression profiling was performed based on the nCounter expression panel analysis (Nanostring Technologies, Seattle, WA, USA) as detailed in [2]. The nCounter^®^ expression panel can identify 577 miRNAs in a single run using sensitive barcodes without the need for reverse transcription. These highly sensitive and specific barcodes can detect even very small amounts of miRNA in the mandibles. The sample preparation for the miRNA nCounter expression panel involves three basic steps: annealing the nCounter miRNA tag, ligation with master mix, and purification of the RNA library to remove the unligated miRNA tags. Consequently, the annealing master mix was prepared by combining annealing buffer, nCounter miRNA tag reagent, and diluted (1:500) miRNA assay controls. Further, the annealing master mix was aliquoted into each tube of the strip, and 100 ng of the total RNA from ten left mandibles from each group with 260/280 and 260/230 ratios of > 2 was added to the respective tubes. The strip tube was transferred in the thermal cycler with the following conditions: 94 °C for 1 min, 65 °C for 1 min, 45 °C for 1 min, and 48 °C for hold. 

Furthermore, following annealing, ligation master mix (polyethylene glycol (PEG) and ligation buffer) was added to all the tubes in the strip tube. The strip tubes were incubated, followed by the addition of ligase to each tube. The ligation was performed with the following conditions: 48 °C for 3 min, 47 °C for 3 min, 46 °C for 3 min, 45 °C for 3 min, 65 °C for 10 min, and 4 °C for hold. In order to separate the unligated tags, a purification step was performed after adding ligation cleanup enzyme to all the tubes and incubating them at 37 °C for 1 h, 70 °C for 10 min, and 4 °C for holding. The RNase-free water was added to each tube in the strip tube, and the specimen was ready for hybridization with the nCounter reporter and capture probes. After denaturation, an aliquot from the miRNA sample preparation tube was taken along with the miRNA reporter code, hybridization buffer, and miRNA capture probe. The strip tubes were incubated in the thermal cycler, and the specimens were immediately processed for post-hybridization with the nCounter analysis system at the Molecular Pathology Core at the University of Florida. The nCounter^®^ Mouse miRNA Assay Kit v1.5 provided six positive hybridization controls and eight negative control probes to monitor hybridization efficiency. All components and reagents needed for specimen preparation at the preparation station were taken from the nCounter Master Kit (Nanostring Technologies). Twelve specimens per cartridge were processed in a single run and followed by digital analysis, which involved the transfer of the cartridge to the multichannel epifluorescence digital analyzer. A cartridge definition file with a maximum fields of view (FOV) count of 555 per flow cell was taken for digital analysis. The number of images taken per scan corresponded to the number of immobilized reporter probes on the cartridge. A separate Reporter Code Count (RCC) file for each sample containing the count for each probe was downloaded and used for data analysis. 

### 4.5. NanoString Data Analysis

All the raw data obtained from the NanoString analysis was initially subjected to a stringent quality check (QC) to obtain unbiased data. In addition, all the samples passed QC, and no flags were observed. The raw data was imported into nSolver 4.0. Data analysis was done based on the methodology described previously [2]. For KEGG (Kyoto Encyclopedia of Genes and Genomes) pathway analysis, we analyzed the DE miRNAs in 8- and 16-weeks bacterially infected mouse mandibles in the DIANA-miRPath v.3.0 database. Further, all the DE miRNAs were entered using the MIMAT accession number in the DIANA-miRPath database with threshold values of *p* < 0.05 and a false discovery rate (FDR) correction applied to obtain an unbiased empirical distribution. A venn diagram for upregulated and downregulated miRNAs in 8- and 16-weeks of infection was drawn using Venny 2.1 [2]. 

### 4.6. Statistical Analysis

All the data in the graphs were presented as mean + SEM. An ordinary one-way ANOVA with Tukey’s multiple comparison test and a single pooled variance was performed for IgG antibody analysis to identify the statistical significance using Prism 9.4.1 (GraphPad Software, San Diego, CA, USA). An ordinary two-way ANOVA with Tukey’s multiple comparison test and a single pooled variance was performed for ABR measurements. Identification of significant differential gene expression was done based on two-tailed t-tests on the log-transformed normalized data that assumed unequal variance. The distribution of the t-statistics was calculated using the Welch-Satterthwaite equation for the degrees of freedom to estimate the 95% confidence intervals for the identified DE of miRNA between groups. A *p*-value of <0.05 was considered statistically significant [2].

## 5. Conclusions

This is the first monobacterial in vivo study that reports the altered global miRNA kinetics in experimental periodontitis with *P. gingivalis.* The study highlights the transient expression of miRNA in mouse gingival tissues, which depends on the duration of *P. gingivalis* infection. In addition, miR-103 and miR-30d were commonly upregulated at both time points and were unique. As miR-103 and -30d are also shown to be upregulated in atherosclerosis and diabetes, these two miRNAs may be considered potential biomarkers for PD progression as well as establishing a miRNA link between PD and systemic diseases. Four upregulated miRNAs (miR-31, miR-125b, miR-15a, and miR-195) were also identified in gingival tissues of periodontitis patients, indicating preclinical in vivo miRs data corroborates with clinical PD miRs data. In-depth analysis of the identified DE miRNAs (miR-103, miR-30d, miR-31, miR-125b, miR-15a, and miR-195) using miRNA-specific knockout mouse models will provide additional roles on the functions of these miRNAs, miRNAs’ links with PD, and a better understanding of pathobiology.

## Figures and Tables

**Figure 1 ijms-24-02327-f001:**
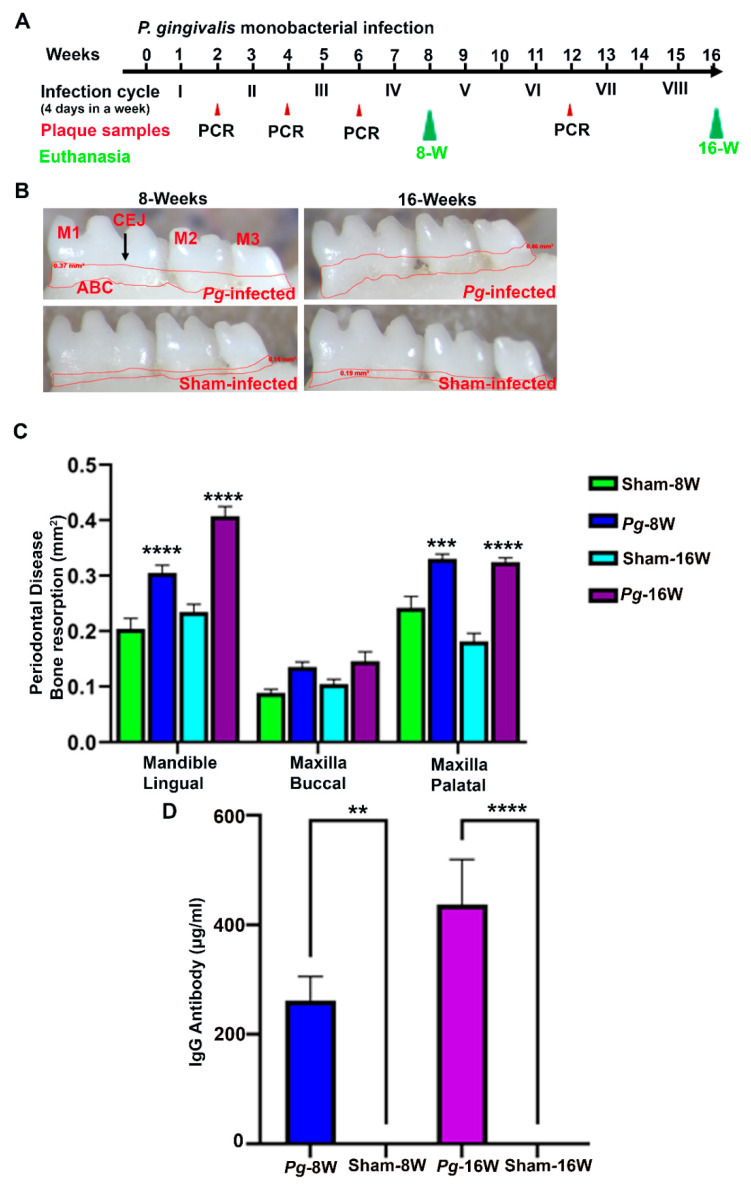
*P. gingivalis* infection significantly induced ABR and elicited an IgG immune response in mice. (**A**). Schematic diagram of the experimental design depicting the *P. gingivalis* monobacterial infection (4 days per week on every alternate week), and plaque sampling for PCR and euthanasia. (**B**). Representative images showing horizontal ABR (mandible lingual view) of *P. gingivalis*-infected and sham-infected mice with the area of bone resorption outlined from the alveolar bone crest (ABC) to the cementoenamel junction (CEJ). (**C**). Morphometric analysis of the mandible and maxillary ABR in mice. A significant increase in ABR was observed in *P. gingivalis*-infected mice compared to sham-infected mice at 8- and 16-weeks. (*** *p* < 0.001, **** *p <* 0.0001; two-way ANOVA). (**D**) Serum IgG immune response against the *P. gingivalis* was also significantly higher in 8-weeks (** *p* < 0.05) and 16-weeks (**** *p* < 0.0001; Ordinary one-way ANOVA). Data points and error bars are mean± SEM (*n* = 10).

**Figure 2 ijms-24-02327-f002:**
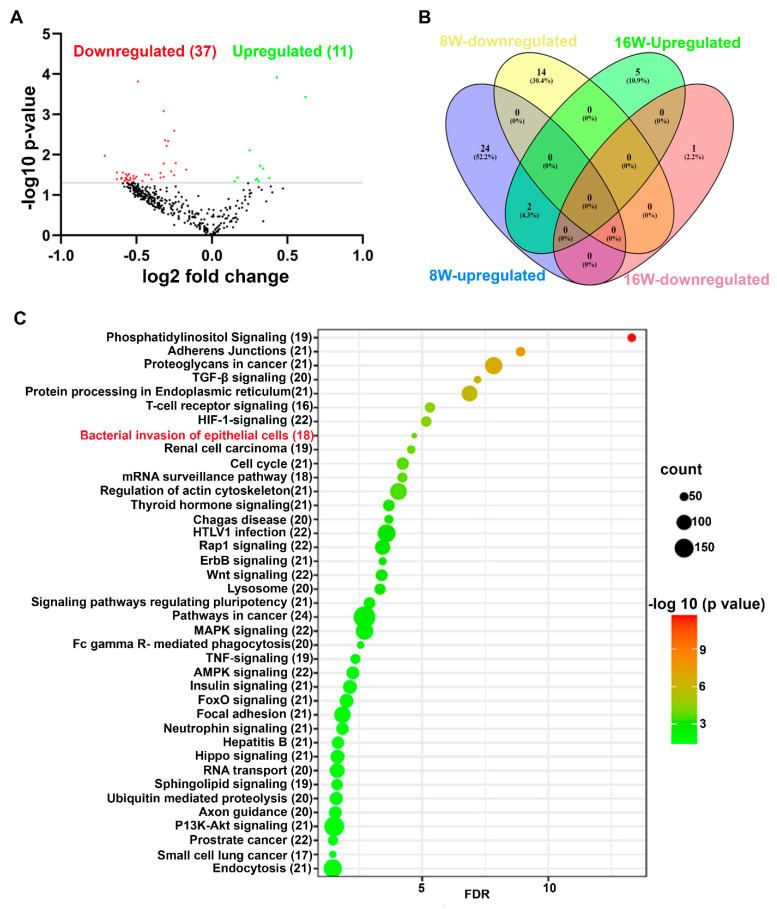
Differentially expressed (DE) miRNAs in *P. gingivalis*-infected mandibles (8- and 16-weeks). (**A**) The volcano plot depicts the upregulated (green) and downregulated (red) miRNAs that showed a fold difference of ±1.1 with *p*-value of < 0.05. The log2 fold change is on the *x*-axis, and the negative log of the *p*-value is on the *y*-axis. The black dots represent the miRNAs that do not pass the filter parameters. Eleven significant upregulated miRs and 37 downregulated miRs were identified in 16-weeks of *P. gingivalis*-infected mice compared to 8-weeks *P. gingivalis*-infected mice (*n* = 10). (**B**) Venn diagram analysis illustrates the distribution of DE miRs in 8-week and 16-weeks infections with *P. gingivalis*. This analysis shows that two miRNAs, miRNA-103 and miR-30d, were commonly upregulated in both 8- and 16-weeks infected mice. All other upregulated and downregulated miRNAs were unique to the 8- and 16-weeks infected mice. (**C**) Predicted functional pathway analysis of DE miRNAs from *P. gingivalis*-infected mandibles. Bubble Plot of KEGG analysis on predicted target genes of DE miRNAs in *P. gingivalis*-infected mice compared to sham-infected mice. The KEGG pathways are displayed on the y-axis, and the x-axis represents the false discovery rate (FDR), which means the probability of false positives in all tests. The size and color of the dots represent the number of predicted genes and corresponding *p*-value, respectively. Eighteen DE miRs were shown to be involved in the bacterial invasion of epithelial cells.

**Table 1 ijms-24-02327-t001:** Gingival plaque samples positive for *P*. *gingivalis* gDNA by PCR.

Group/Bacteria/Infection (Weeks)	Positive Gingival Plaque Samples (*n* = 10) ^a^			
	2 Weeks	4 Weeks	6 Weeks	12 Weeks
Group I/ *P. gingivalis* FDC 381 [8 weeks]	8/10 ^a^	10/10	NC	---
Group II/ *P. gingivalis* FDC 381 [16 weeks]	10/10	10/10	NC	10/10
Group III/ Sham-infection [8 weeks]	0/10	NC	NC	---
Group IV/ Sham-infection [16 weeks]	0/10	NC	NC	0/10

^a^ Total numbers of gingival plaque samples that were collected after infections (2, 4, 6, and 12 weeks) and positive infections were determined by PCR analysis. NC—not collected to allow bacterial biofilm to adhere to gingival surface, invade epithelial cells, and multiply. ^a^ the first value corresponds to the number of mice that tested positive for the respective genomic DNA and the second value corresponds to the total number of mice in the group.

**Table 2 ijms-24-02327-t002:** Differentially expressed miRNAs during 8- and 16-weeks *P. gingivalis* infection in mice.

Weeks/Infection/Sex	Upregulated miRNAs (*p* < 0.05)	Downregulated miRNAs (*p* < 0.05)
8W- *P. gingivalis* infected Vs. 8W- Sham infection (*n* = 10)	26 (miR-103, miR-30d)	14
8W- *P. gingivalis* infected Female Vs Male (*n* = 5)	3	2
16W- *P. gingivalis* infected Vs. 16W- Sham infection (*n* = 10)	7 (miR-103, miR-30d)	1
16W- *P. gingivalis* infected Female Vs Male (*n* = 5)	39	9
8W- *P. gingivalis* infected Vs. 16W- *P. gingivalis* infected	11	37

**Table 3 ijms-24-02327-t003:** Upregulated miRNAs, molecular functions, and target genes.

**Upregulated miRNAs in 8-Weeks of** * **P. gingivalis** * **Infection**				
**miRNAs**	**Fold Change**	***p*-Value**	**Reported Functions**	**Number of Target Genes**
miR-804	1.57	0.009465	Not identified	--
miR-690	1.53	0.007031	Inflammation and endoplasmic reticulum stress in obese animal models [10].	157 (*Srrm2, Fkbp1a, Glu1, Brd8, Cyb5r1, Slc16a12, Cdip1, Tob, Megf9, Hc*)
miR-1224	1.4	0.013375	Ischemic stroke by acting as negative regulator of Natural Killer cells [11].	--
miR-31	1.37	0.034075	Tumor suppressor in many cancers such as Ovarian, HCC, Prostate, Glioma, and Breast Cancer [12].	1176 (*Mul1, Nipsnap3b, Cyc1, Srrm2, Fubp1, Wipi2, Gcnt4, Kat6b, Arid3a, Bace1*)
miR-133b	1.36	0.03893	Regulates TLR component of IRAK-1 and promotes phagocytosis [13]	1430 (*Med13, Rrp8, Cnp, Srrm2, Wipi2, Sh3pxd2b, Dlc1, Tgfb2, Rsbn1l, Kat6b*)
miR-185	1.32	0.000534	Regulates angiogenesis in Prostate Cancer [14]	332
miR-19b	1.31	0.000122	Attenuates TNF-induced endothelial cell apoptosis in coronary heart disease [15]	1901
miR-22	1.31	0.009126	Regulates endothelial inflammation, tissue injury and angiogenesis [16].	2150
miR-193	1.29	0.006863	Attenuates myocardial injury of mice with sepsis [17].	0
miR-1198	1.27	0.001943	Reported to be downregulated in Juvenile cataracts mouse model [18].	3
miR-154	1.27	0.04589	Mediates allergic reactions that include passive cutaneous and systemic anaphylaxis [19].	541
miR-152	1.26	0.031135	Inhibition of cell proliferation and progression in breast cancer, glioma and gastric cancer [20].	1115
miR-423-3p	1.26	0.044217	Breast cancer invasion by activating NF-κB signaling pathway [21].	4
miR-125b-5p	1.24	0.02166	Associated with acute Ischemic stroke [22].	1624
miR-191	1.23	0.013163	Promotes ischemic brain injury by inhibiting angiogenesis [23].	129
miR-107	1.22	0.022686	Insulin resistance in Type II Diabetes, Inflammation and Obesity [24].	311
miR-103	1.21	0.004192	Highly expressed in ApoE^-/-^ mice and linked with atherosclerosis [25].	2
miR-322	1.19	0.019203	Over expressed in *Hdac3* KO mouse epicardial cells and hearts [26].	2761
miR-30d	1.17	0.042862	Promotes cardiomyocyte pyroptosis in diabetic cardiomyopathy [27].	10
miR-301a	1.15	0.041072	Promotes oxidative stress, inflammation and apoptosis in arteriosclerotic cardiovascular disease [28].	1220
miR-15a	1.15	0.046356	Negative regulator of postischemic cerebral angiogenesis and long-term neurological recovery [29].	1368
miR-30c	1.15	0.049249	Regulate macrophage mediated inflammation and pro-atherosclerosis pathways [30].	1297
miR-30e	1.13	0.00518	Protective role against neuronal deficit and inflammation in intracerebral hemorrhage [31].	1324
miR-151-5p	1.13	0.02108	Breast cancer metastasis [32]	113
miR-28	1.13	0.041907	Oncogene that promotes human glioblastoma cell growth [33].	0
miR-151-3p	1.11	0.030216	Regulates inflammation and apoptosis [34].	81
**Upregulated miRNAs in 16-Weeks of *P. gingivalis* Infection**				
**miRNAs**	**Fold Change**	***p*-Value**	**Reported Functions**	**Number of Target Genes**
miR-30d	1.11	0.046212	Promotes cardiomyocyte pyroptosis in diabetic cardiomyopathy [27].	10 *Pik3r1, Ctnna1, Cblb, Sept8, Was1, Sept2, Rac1, Crk, Itgb1, Pik3cd*
miR-103	1.13	0.044262	Highly expressed in ApoE^-/-^ mice and linked with atherosclerosis [25].	2 (*Cltc, Rac1*)
miR-145	1.14	0.018223	Oncogene that enhances migration and invasion of Hepatocellular carcinoma [35].	0
miR-195	1.18	0.028636	Modulates cell proliferation in colorectal cancers [36].	0
miR-24	1.19	0.020169	Overexpression of miR-24 reduced the effect of *S. aureus* in osteomyelitis patients [37].	359 (*Srrm2, Rsbn1l, Axin1, Sla, Bc030336, Mef2, Fyco1, Wipf1, Kidins220, Mhc*)
miR-365	1.22	0.045178	Highly expressed in patients with Osteoarthritis [38].	0
miR-99b	1.24	0.001738	Predicts clinical outcome of osteosarcoma.	4

Details of the target genes were given for the top five significantly expressed miRNAs in both 8-week and 16-week infections.

## Data Availability

The raw data and other related data in the manuscript are available from the corresponding author, L.K., upon reasonable request.

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
