# Peer review of "Specific microRNA Signature Kinetics in Porphyromonas gingivalis-Induced Periodontitis"

_ijms, 2023, doi:10.3390/ijms24032327_

Round 1

Reviewer 1 Report

i would like to thank the authors for this very good and interesting paper.

However, i have some minors remarks :

THE FIRST ONE AND MOST IMPORTANT : i don't understand why the Materials and Methods is placed in the section 4, after the results and the discussion...

The Materials and methods has to be in section 2 after the introduction

some minor remarks concerning the text /

PAGE 2 LINE 69 : even if you explain the meaning of DE in the abstract, you may do it again here 

PAGE 9 tABLE 3 :  THE REFERENCES ARE MISSING FOR miR-1198 and miR-152

PAGE 13 LINE 394 : please add "per group" at (n=10)

Author Response

Reviewer #1

Query 1: The first one and most important: i don't understand why the Materials and Methods is placed in the section 4, after the results and the discussion...

Authors’ response:

This IJMS manuscript was prepared as per the IJMS journal format (Microsoft word template) and it is clearly asked the authors to include materials and methods in section 4 after the results in section 2 followed by discussion in section 3.

Query 2: PAGE 2 LINE 69: even if you explain the meaning of DE in the abstract, you may do it again here

Authors’ response:

As per the reviewer’s suggestion, DE is expanded as Differential expression. Please refer line #70.

Query 3: PAGE 9 TABLE 3: THE REFERENCES ARE MISSING FOR miR-1198 and miR-152

Authors’ response:

References for both miR-1198 and miR-152 was added in the revised manuscript. Please refer reference# 18 and 20 and it was also updated in Table 3.

Query 4: PAGE 13 LINE 394: please add "per group" at (n=10)

Authors’ response:

As per the reviewer’s suggestion, “mice/per group; 5 males and 5 females” was included. Please refer line 398 and also in line #429.

Reviewer #2

Query 1: Discuss results that showed no significant differences in ABR was observed in 8-weeks and 16-weeks P. gingivalis infected mice. 

Authors’ response:

We feel reviewer is confused regarding alveolar bone resorption (ABR): We have stated as follows (Line 103-105). Furthermore, gingival infection of P. gingivalis significantly induced horizontal ABR. ABR was significantly higher in mandible lingual and maxilla palatal (p<0.0001) for both the 8-weeks and 16-weeks P. gingivalis-infected mice (Figure. 1B-C).

Query 2: miR-103, 146 and miR-30d were upregulated in the current model, yet both were not shown to be up-regulated in human may question model used Thus, the results must be validated by qPCR, or tune down its importance to be used as a marker in the discussion and conclusion.

Authors’ response:

We used high-throughput nCounter® miRNA Expression Panels from Nanostring Technologies, to effectively identify the differentially expressed miRNAs in 8-weeks and 16-weeks P. gingivalis infected mice. Nanostring nCounter® analysis can identify 577 miRNAs in any sample, and by using molecular barcodes, it can detect even a low number of miRNAs without the need for reverse transcription or amplification. Hence, the probability to introduce artifacts is limited as it does not involve cDNA conversion as like in Real-Time PCR. Hence, there is no need to validate the NanaoString data using stem-loop Real-Time PCR. To verify the sensitivity and specificity, we have done the q-PCR analysis of the inflammatory miRNA 146a and found that both the q-PCR and NanoString data were similar. Based on this analysis, we decided not to validate any NanoString data as it is highly specific and accurate. Please refer miR-146a Q-PCR data (below).

Figure: Real-Time PCR analysis of mice mandibles from the pilot study. miR-146a expression was quantified from the mice mandibles using the traditional Q-PCR approach (Stem-loop Real-Time PCR) in which normalization was done using the SnoRNA202. Amplifications were done in triplicate for each sample (n=6). No significant changes in the mir-146a was observed in mice infected with partial human mouth microbes (PAHMM) compared to sham-infected mice. We also observed the similar expression for miR-146a in NanoString analysis with the same samples. Hence, it was confirmed that the NanoString analysis of the mice mandibles were correct and the data were consistent.

Query 3: One limitation that needs to be mentioned (may be difficult) that the current study did not utilize saliva fluids

Authors’ response:

We thank the reviewer for the suggestion. We will design our next study with the inclusions of saliva fluids. We have mentioned this limitation in the revised manuscript. Please refer line# 385-386.

Query 4: The authors should recognize that NanoString assay is likely to have lower sensitivity and the RNA input was not mentioned in the materials.

Authors’ response:

We have added detailed description of RNA isolation, concentration, purity determination and RNA quantification for more clarity (Line # 431-444). 100 ng of the total RNA from all the mandibles from each group (maximum 3-µL volume) were taken for the present study. In addition, we have added detailed description of sample preparation for miRNA nCounter expression panel that involves three basic steps annealing the nCounter miRNA tag, ligation with master mix, and purification of RNA library to remove the unligated miRNA tags (Line # 452-481). As mentioned earlier, NanoString is highly specific and it can detect even a low number of miRNAs without the need for reverse transcription or amplification.

Query 5: It was not clear number of female and male mice were used and if there was differences in DE between them.

Authors’ response:

We used both males (n=5) and females (n=5) and clearly found the differences in DE between them. Please refer Table 2. We have clearly shown this difference in Table 2. The details were also mentioned in line#117-118 and 121-122.

Reviewer 2 Report

The current manuscript highlights the importance of using miRNAs as a novel class of highly sensitive and specific biomarkers for periodontitis.

Minor concerns:

1. Discuss results that showed no significant differences in ABR was observed in 8-weeks and 16-weeks P. gingivalis infected mice. 

2.miR-103, 146 and miR-30d were upregulated in the current model, yet both were not shown to be up-regulated in human may question model used Thus, the results must be validated by qPCR, or tune down its importance to be used as a marker in the discussion and conclusion.

3. One limitation that needs to be mention (may be difficult) that the current study did not utilize saliva fluids.

4. The authors should recognize that NanoString assay is likely to have lower sensitivity and the RNA input was not mentioned in the materials.

5. It was not clear number of female and male mice were used and if there was differences in DE between them.

Thank you

Author Response

Reviewer #2

Query 1: Discuss results that showed no significant differences in ABR was observed in 8-weeks and 16-weeks P. gingivalis infected mice. 

Authors’ response:

We feel reviewer is confused regarding alveolar bone resorption (ABR): We have stated as follows (Line 103-105). Furthermore, gingival infection of P. gingivalis significantly induced horizontal ABR. ABR was significantly higher in mandible lingual and maxilla palatal (p<0.0001) for both the 8-weeks and 16-weeks P. gingivalis-infected mice (Figure. 1B-C).

Query 2: miR-103, 146 and miR-30d were upregulated in the current model, yet both were not shown to be up-regulated in human may question model used Thus, the results must be validated by qPCR, or tune down its importance to be used as a marker in the discussion and conclusion.

Authors’ response:

We used high-throughput nCounter® miRNA Expression Panels from Nanostring Technologies, to effectively identify the differentially expressed miRNAs in 8-weeks and 16-weeks P. gingivalis infected mice. Nanostring nCounter® analysis can identify 577 miRNAs in any sample, and by using molecular barcodes, it can detect even a low number of miRNAs without the need for reverse transcription or amplification. Hence, the probability to introduce artifacts is limited as it does not involve cDNA conversion as like in Real-Time PCR. Hence, there is no need to validate the NanaoString data using stem-loop Real-Time PCR. To verify the sensitivity and specificity, we have done the q-PCR analysis of the inflammatory miRNA 146a and found that both the q-PCR and NanoString data were similar. Based on this analysis, we decided not to validate any NanoString data as it is highly specific and accurate. Please refer miR-146a Q-PCR data (below).

Figure: Real-Time PCR analysis of mice mandibles from the pilot study. miR-146a expression was quantified from the mice mandibles using the traditional Q-PCR approach (Stem-loop Real-Time PCR) in which normalization was done using the SnoRNA202. Amplifications were done in triplicate for each sample (n=6). No significant changes in the mir-146a was observed in mice infected with partial human mouth microbes (PAHMM) compared to sham-infected mice. We also observed the similar expression for miR-146a in NanoString analysis with the same samples. Hence, it was confirmed that the NanoString analysis of the mice mandibles were correct and the data were consistent.

Query 3: One limitation that needs to be mentioned (may be difficult) that the current study did not utilize saliva fluids

Authors’ response:

We thank the reviewer for the suggestion. We will design our next study with the inclusions of saliva fluids. We have mentioned this limitation in the revised manuscript. Please refer line# 385-386.

Query 4: The authors should recognize that NanoString assay is likely to have lower sensitivity and the RNA input was not mentioned in the materials.

Authors’ response:

We have added detailed description of RNA isolation, concentration, purity determination and RNA quantification for more clarity (Line # 431-444). 100 ng of the total RNA from all the mandibles from each group (maximum 3-µL volume) were taken for the present study. In addition, we have added detailed description of sample preparation for miRNA nCounter expression panel that involves three basic steps annealing the nCounter miRNA tag, ligation with master mix, and purification of RNA library to remove the unligated miRNA tags (Line # 452-481). As mentioned earlier, NanoString is highly specific and it can detect even a low number of miRNAs without the need for reverse transcription or amplification.

Query 5: It was not clear number of female and male mice were used and if there was differences in DE between them.

Authors’ response:

We used both males (n=5) and females (n=5) and clearly found the differences in DE between them. Please refer Table 2. We have clearly shown this difference in Table 2. The details were also mentioned in line#117-118 and 121-122.
